# Dissection of innate-immune-ligand- and interferon-protein-mediated transcriptional responses in human THP1 cell states
Lodoe Lama, Pavel Morozov ®[2] , Aitor Garzia & Thomas Tuschl ® ✉

Interferon regulatory factors (IRFs) are essential for transcription of interferons (IFNs), interferon-stimulated genes (ISGs), and pro-inflammatory cytokines. We profile the transcriptome of human monocyte THP1 cells challenged with cGAMP, LPS, or IFNB1 protein as a function of knockout (KO) or overexpression (OE) of IRFs or KO of IFNAR2. We define distinct gene expression groups, reflecting the transcription factors responsible for their induction including subgroups activated by more than one pathway or feed-forward regulation. We compare IRF3- and IRF7-induced gene signatures and note the strong direct induction of a subset of antiviral-acting ISGs by IRF3 or IRF7. LPS treatment induces NF-κB responses in monocyte and macrophage state cells, however, IFNs and ISGs are only co-induced in the macrophage state requiring IRF3. IRF1, IRF2, IRF5, and IRF8 are largely dispensable for IFN-stimulated or innate-immune-mediated gene induction. This study provides a valuable resource for dissecting complex inflammatory gene signatures and their underlying transcription factors thereby anticipating the effects of selectively drugging the underlying pathways.

Interferons (IFNs), chemokines, and pro-inflammatory cytokines are secreted signaling proteins central to innate and adaptive immunity for defense against pathogens[1]. These signaling proteins are induced upon activation of pattern recognition receptors (PRRs) for pathogen-associated molecular patterns (PAMPs)[2] and damage-associated molecular patterns (DAMPs)[3] of compromised host cell. PRRs include toll-like receptors (TLRs)[4], retinoic-acid-inducible-gene-I- (RIG-I-) like-receptors (RLRs)[5], and cyclic GMP-AMP synthase (CGAS)[6]. These PRRs, when activated, engage various adaptor proteins and kinases leading to phosphorylation and activation of members of various families of transcription factors including interferon regulatory factors (IRFs) and NF-κB complexes[7]. The expressed IFN and cytokine proteins engage in autocrine and paracrine signaling triggering additional transcriptional responses, including the production of hundreds of IFN-stimulated genes (ISGs), thereby complicating deconvolution of the initiating inflammatory response. Cells exposed only to type I IFN protein do not transcribe interferon genes. A feed-forward induction of type I IFN genes requires long-lasting or repeated innate immune stimulation ensuing IFN protein synthesis[8,9].

IFNs and ISGs lead into a cellular antiviral state suppressing intracellular pathogen replication. Co-induced chemokines and pro-inflammatory cytokines alert the adaptive pathogen-selective immune system to initiate T and B cell development for clearance of infected cells and prevention of new infection, respectively[10].

IFNs are categorized into three types based on the cell surface receptors they engage[1]. Relevant to innate responses are type I and type III IFN genes[11]. Type I includes IFNβ (*IFNB1*), IFNε (*IFNE*), IFNκ (*IFNK*), IFNω (*IFNW*), and 13 IFNα (*IFNA1*, *2*, etc.) and type III includes IFNλ1 (*IFNL1*), IFNλ2 (*IFNL2*), and IFNλ3 (*IFNL3*). Type I IFNs bind the IFNAR1-IFNAR2 cell-surface-localized heterodimeric receptor while type III IFNs bind IFNLR1-IL10RB. IFNAR1 and IFNAR2 are broadly expressed while IFNLR1 and IL10RB are enriched in epithelial and immune cells, respectively. Engagement of type I or type III IFN receptors activates JAK1 and TYK2 kinases which in turn phosphorylate STAT1 and STAT2 transcription factors. Phosphorylated STATs form various transcription factor complexes[12,13] responsible for induction of ISGs.

Concomitant with IFNs is the induction of CXCL (CXCL10, CXCL11)[14] and CCL (CCL3, CCL4, CCL5) family chemokines[15,16]. These chemokines mediate their effects by binding to their respective CXCR or CCR receptors present on target cells, initiating diverse intracellular signaling cascades[17]. Activation of the NF-κB pathway leads to the production

Laboratory for RNA Molecular Biology, The Rockefeller University, New York, NY, USA. ✉e-mail: ttuschl@rockefeller.edu

https://doi.org/10.1038/s42003-025-09343-7   **Article**

of pro-inflammatory cytokines (IL1B, IL8, TNF)[18], a subset of which needs proteolytic processing for their activation and pyroptosis for their release and intercellular signaling[19].

The human genome encodes nine members of the IRF family, defined by an N-terminal DNA-binding domain that recognizes GAAA dsDNA sequence motifs[20]. IRF1, 3, 5, 7, and 8 have been implicated in IFN expression. IRF1 was the first member identified[21] though subsequent studies in IRF1 knockout (KO) mice[22] implicated IRF3 and IRF7 as major IFN transcription factors active upon viral infection[23]. IRF3 expression is constitutive, while *IRF1*, *IRF7*, and *IRF9* are also ISGs, IRF1 and IRF7 being responsible for feed-forward IFN gene transcription during persistent innate immune activation[24–26]. IRF3 and IRF7 vary in dsDNA-binding specificity, with IRF7 being able to induce expression of all type I IFN genes, while IRF3 is selective for IFNB1[25]. It should be noted that plasmacytoid dendritic cells (pDCs), which are known for their ability of massive IFN production, show constitutively high expression of IRF7[27]. IRF5 is implicated in TLR7- and TLR8-dependent IFN induction[28,29]. IRF8, which is predominantly expressed in human blood monocytes, constitutively binds to the IFNB1 promoter and facilitates rapid IRF3-dependent transcription of IFNB1 upon viral infection[30]. IRF9 participates in downstream IFN-receptor-mediated activation of ISGs by forming the phosphorylated STAT1-STAT2-IRF9 ISGF3 complex[13].

The subcellular localization of the PRRs is also regulated and differentially impacts the various inflammatory pathways. In monocytes, TLR4 is predominantly located on the plasma membrane[31] and mostly signals via the TIRAP-MYD88 adapter complex[32] leading to downstream activation of NF-κB pathway. As shown in bone-marrow-derived macrophages, the selectively co-expressed GPI-anchored protein CD14[33] facilitates LPS-induced TLR4 endocytosis and engagement of the TRAM-TRIF pairs of adapter proteins[34] leading to co-induction of IRF3-dependent IFN and ISG expression aside from the TIRAP-MYD88-mediated NF-κB gene activation[35].

The role of IRFs as transcriptional regulators of innate immune response has been established, particularly for the induction of IFNs and ISGs[36]. However, a recent review[37] underscores a considerably more complex innate immune signaling processes extending beyond a sequential transcriptional response for IRF-induced IFNs and IFN-protein-induced ISGs. For instance, IRF3 had been implicated in direct upregulation of the prototypical antiviral-acting ISGs *IFIT1*, *2*, *3*, and *ISG15* independent of IFN receptor signaling[38].

The human leukemia-derived monocyte THP1[39] cell line is widely used in immune modulation studies[40] encompassing inflammatory responses[41], host-pathogen interaction mechanisms[42–44], phagocytosis and antigen presentation mechanisms[45], toxicology in drug development[46], and macrophage polarization[47]. Treatment of THP1 monocytes with phorbol 12-myristate 13-acetate (PMA), a mimic of the second messenger diacylglycerol, leads to their terminal differentiation into adherent macrophage-like cells of the M0 state with remarkable phenotypic and functional characteristics of human monocyte-derived macrophages derived from peripheral blood mononuclear cell[48,49]. M0 macrophages can be further polarized into M1- or M2-type "activated" macrophages by treatment with IFNγ-LPS or interleukin-4 (IL-4), respectively[47]. Given that genes involved in innate immunity are non-essential in cultured THP1 cells, their roles are readily explored using CRISPR-based gene KO. Individual studies in THP1 cells utilizing KO of various PRR genes[50,51], IRF3[52], IFNAR receptor[53], and IRF9[54] have helped to establish and validate their roles in IFNs and ISG induction. However, a comprehensive study dissecting transcriptional activation by individual IRF family members in innate immune sensing and downstream IFN-protein-mediated induction of ISGs and other inflammatory pathways is still lacking. Notably, only a single study compared the role of six IRF KOs in human alveolar epithelial A549 cell line by monitoring a small subset of IFN, cytokine, and ISG mRNAs by RT-qPCR[55].

Here we characterized THP1 wild-type and KO cells by transcriptome-wide RNA-seq to classify innate immune responses and their transcriptional pathways as a function of IRFs and IFNAR2. We defined 1) CGAS-STING1-induced genes dependent on IRF3 or IRF7, including type I IFNs, 2) a subset of ISGs that are also directly induced by IRF3 or IRF7, 3) IRF9-dependent and independent ISGs, and 4) NF-κB-pathway-induced genes. Activation of the CGAS-STING1 pathway using cGAMP does not activate the NF-κB pathway. However, LPS treatment activated the NF-κB response in monocyte and macrophage THP1 cell states, and selective co-induction of type I IFNs and ISGs in macrophages only.

## Results

### THP1 cells as model system to dissect innate immunity transcriptional responses

THP1 cells expressed PRRs *CGAS*, *DDX58/RIG-I*, *IFIH1/MDA5*, *TLR2*, and *TLR4* (Supplementary Data 1, Supplementary Fig. 1A). They also expressed intermediary adapters *STING1*, *MAVS*, *MYD88*, *TIRAP*, *TRAM*, *TRIF*, kinases *TBK1*, *IKBKB*, *IKBKE*, and *CHUK*, and seven of the nine *IRFs* and *NF-κB* transcription factors essential for PRR-mediated transcriptional response including activation of IFN genes. Additionally, THP1 cells expressed *IFNAR1* and *2*, *TYK2*, *STAT1*, *STAT2*, and *IRF9* required for induction of ISGs. The *IFNLR1* receptor required for type III IFN-mediated induction of ISGs was not expressed. *CD14* required for internalization of TLR4[33] was only expressed in macrophage THP1 cells.

A time course of cGAMP treatment using digitonin-permeabilized THP1 monocytes revealed a biphasic posttranslational response (Supplementary Fig. 1B). The first phase of phosphorylation of STING1, TBK1, and IRF3 S386 was detectable at 0.5 h, plateaued at 2.5 h, and persisted up to 4.5 h post treatment. The second phase, marked by phosphorylation of STAT1 Y701, emerged 2.5 h post treatment. By contrast, direct IFNB1 protein treatment of THP1 cells showed a less than 30 min onset of STAT1 Y701 phosphorylation and lasted for about 4 h (Supplementary Fig. 1C). The first phase following cGAMP exposure corresponded to the innate immune sensing as exemplified by IRF3-dependent IFNB1 protein synthesis, the second phase was caused by IFNAR signaling due to IFNB1 protein binding. The 2-h delay in STAT1 Y701 phosphorylation by cGAMP treatment versus direct IFNB1 protein exposure represents the time needed for IFNB1 gene transcription and translation. Similarly, increased protein levels of ISGs, such as IRF9, were only noticeable 3 h after IFNB1 protein treatment, and further increased and lasted for at least 8 h.

*IFNB1* and *IFIT1* mRNAs were induced 1 h post treatment (Supplementary Fig. 1D) but increased further and plateaued at 2 h. *IFIT1*, but not *IFNB1*, was also induced 2 h post IFNB1 protein treatment (Supplementary Fig. 1E). Therefore, *IFIT1* is a direct target of innate immune activation as well as an ISG[38]. A 2 h time point after cGAMP delivery by lipid-nanoparticle-based transfection or digitonin-mediated membrane permeabilization showed 24- or 260-fold increased *IFNB1* mRNA abundance by RT-qPCR, respectively, compared to plain addition of cGAMP into the culture medium (Supplementary Fig. 1F). We selected digitonin-mediated cGAMP delivery for this study because it showed the highest level of *IFNB1* mRNA induction.

LPS treatment of monocyte THP1 cells led to selective activation of the NF-κB pathway 1 h post treatment as monitored by RELA phosphorylation in absence of any IRF3 phosphorylation and at concentrations of up to 1 μg/ml LPS in culture medium (Supplementary Fig. 1G). However, LPS treatment of macrophage THP1 cells additionally induced *IFNB1* mRNA (Supplementary Fig. 1H).

In summary, these time-dependent observations prompted us to select a 2 h time point for RNA-seq analysis of the innate immune responses. This time point captured the peak of early and direct transcriptional responses by innate immune sensing exemplified by *IFNB1* mRNA production. A 2 h time point following innate immune activation also captured the onset of interferon-protein-stimulated gene (ISG) transcription but was still ahead of ISG protein synthesis and therefore free of feed-forward gene regulation. A complete set of ISGs was determined by direct treatment of cells with IFNB1 protein for 2 and 24 h[56,57]. Finally, we included a subset of LPS-treated macrophage THP1 cells to capture the CD14-dependent TLR4-mediated co-induction of IFNs and ISGs.

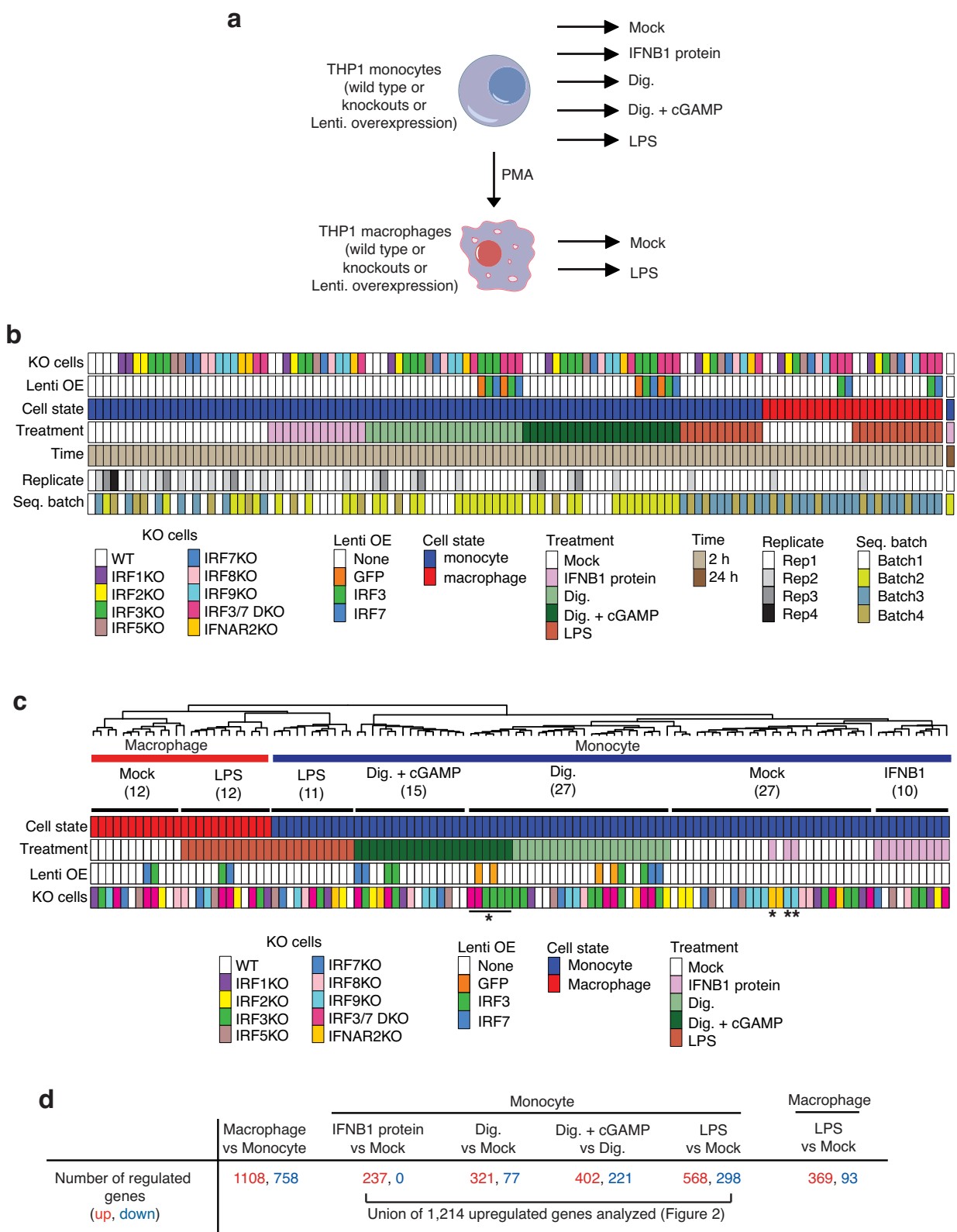

**Fig. 1 | Summary of experimental approach and analysis. a** THP1 cell states and treatments. **b** Sample overview for transcriptome analysis. **c** Organization of samples into seven subgroups by unsupervised clustering as shown in Supplementary Fig. 4. Numbers in parentheses denote the total number of samples included in the DGE analysis for each subgroup. * denotes outlier samples that did not cluster with their expected treatment group. **d** Differentially regulated gene from comparing responses to the various treatments. A total of 1214 upregulated genes in monocytes under various treatments were selected for further analysis as shown in Fig. 2. Dig. digitonin, Lenti OE lentiviral OE of the indicated protein.

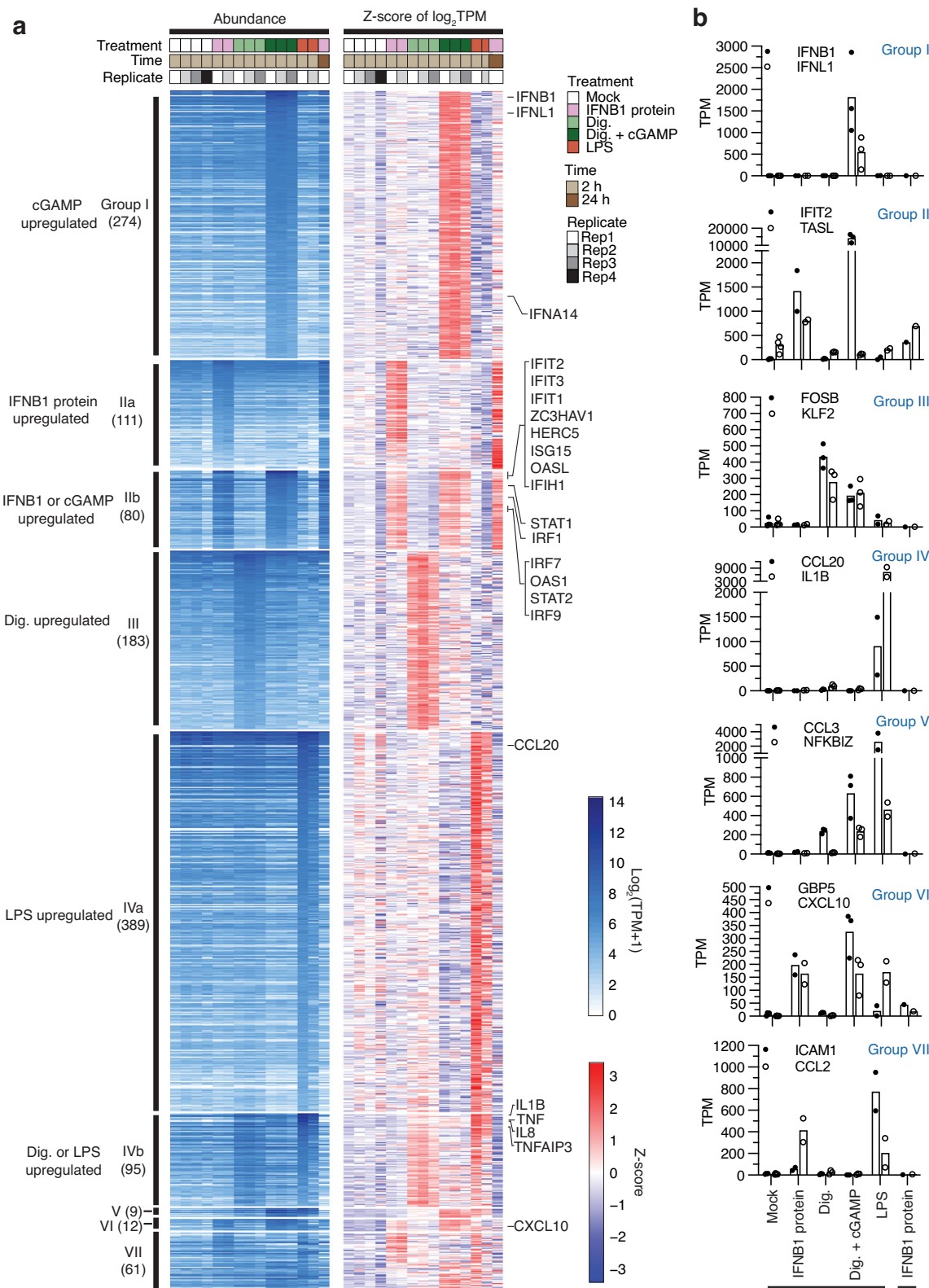

**Fig. 2 | Gene groups upregulated by different treatments in THP1 monocytes.**
**a** Heatmaps displaying either absolute abundance (TPM, left panel) or row-normalized differential expression (z-score, right panel) of upregulated genes under various treatments in parental THP1 monocytes. Genes are grouped based on their unique or overlapping regulation across treatments as labeled on the left side of the heatmap. Genes assigned to various groups are listed in Supplementary Data 3. Each row corresponds to a gene, while each column represents a sample, either untreated or treated with IFNB1 protein, digitonin, digitonin with cGAMP, or LPS. Genes within each group are sorted by decreasing abundance. Numbers in parentheses denote the total number of genes identified per group. Representative genes are indicated by name. The transcript abundance scale also applies to subsequent figures. Dig. digitonin, TPM transcripts per million. **b** Expression levels (TPM) of representative genes from each gene group in parental THP1 monocytes. Each data point represents the TPM value for an individual sample, as listed in Supplementary Data 3, with the mean indicated by the bar.

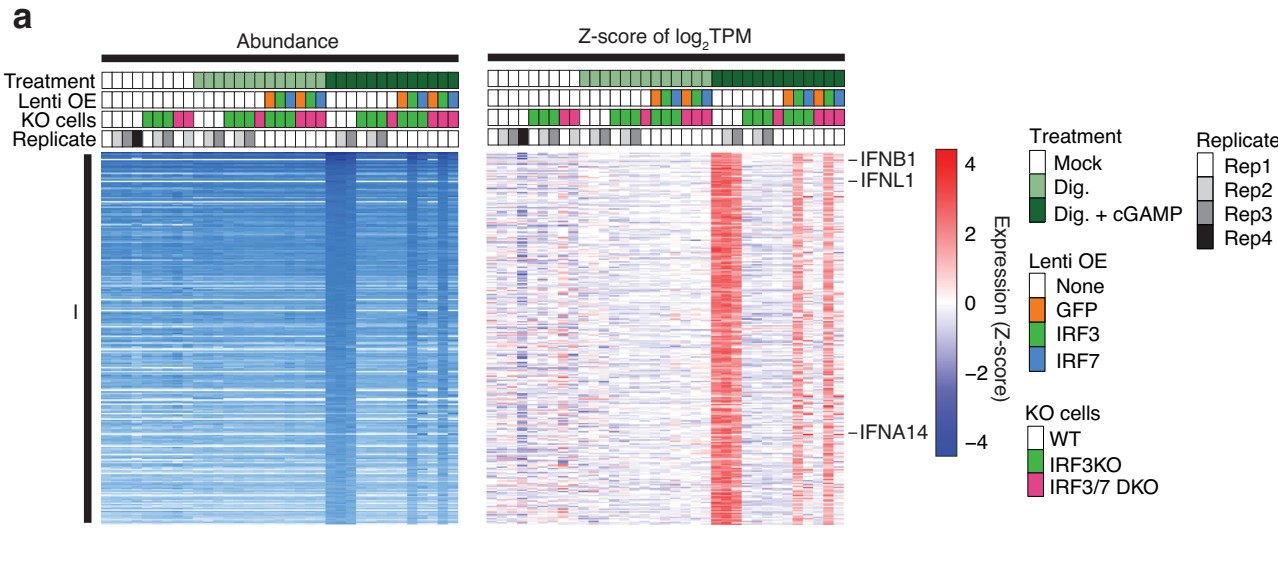

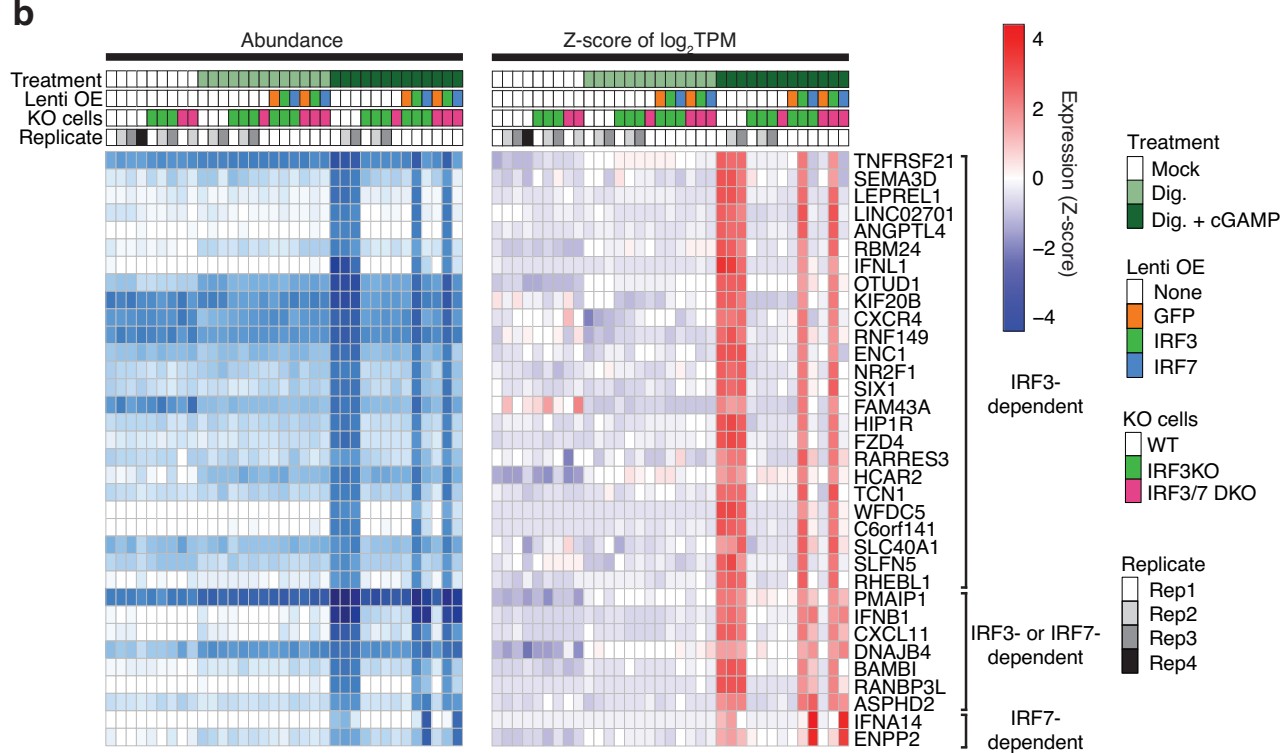

**Fig. 3 | Validation of Group I genes using IRF KO and OE in THP1 monocytes.** **a** Group I genes uniquely induced by cGAMP in wild-type THP1 are impacted by IRF KO or OE. **b** A selected subset of Group I genes demonstrating distinct IRF3- or IRF7-dependent induction in response to cGAMP treatment. A total of 88 Group I genes ($\log_2$FC > 2, TPM > 20) were evaluated, and those showing the following fold induction threshold criteria in OE samples are shown: IRF3-dependent ($\log_2$FC > 2 in IRF3 OE, $\log_2$FC < 2 in IRF7 OE), IRF3- or IRF7-dependent ($\log_2$FC > 2 in IRF3 or IRF7 OE), and IRF7-dependent ($\log_2$FC > 2 in IRF7 OE, $\log_2$FC < 2 in IRF3 OE). Genes uniquely or commonly upregulated by IRF3 or IRF7 are indicated. Only 25 of the 45 identified IRF3-dependent genes are arbitrarily selected and displayed for simplicity. Dig. digitonin, Lenti OE lentiviral OE of the indicated protein.

## Comparative transcriptomic analysis identified IRF-, STAT-, NF-κB-, and stress-response-specific gene sets in THP1 monocytes

To assess the individual contributions of IRF family members to innate immunity transcriptional responses, we created KO lines for IRF1, 2, 3, 5, 7, 8, and 9 including the double KO (DKO) of IRF3 and 7 (Supplementary Figs. 2, 3). We used lentiviral expression of various IRFs to confirm rescue of KO lines. We also generated an IFNAR2 KO line to unmask IRF-induced transcription from IFNB1 protein signaling and ISG expression.

We exposed wild-type and KO THP1 cells to different innate immune stimuli or IFNB1 protein (Fig. 1a) and performed polyA RNA-seq yielding 114 expression profiles (Fig. 1b, Supplementary Data 2). For RNA-seq data analysis, we first conducted pairwise comparisons to identify genes which were ≥4-fold or ≥8-fold regulated by digitonin treatment, immune stimulation or differentiation, respectively, and expressed ≥300 TPM in the induced state, resulting in 360 genes. This gene set was used for unsupervised clustering of all samples and yielded seven sample subgroups

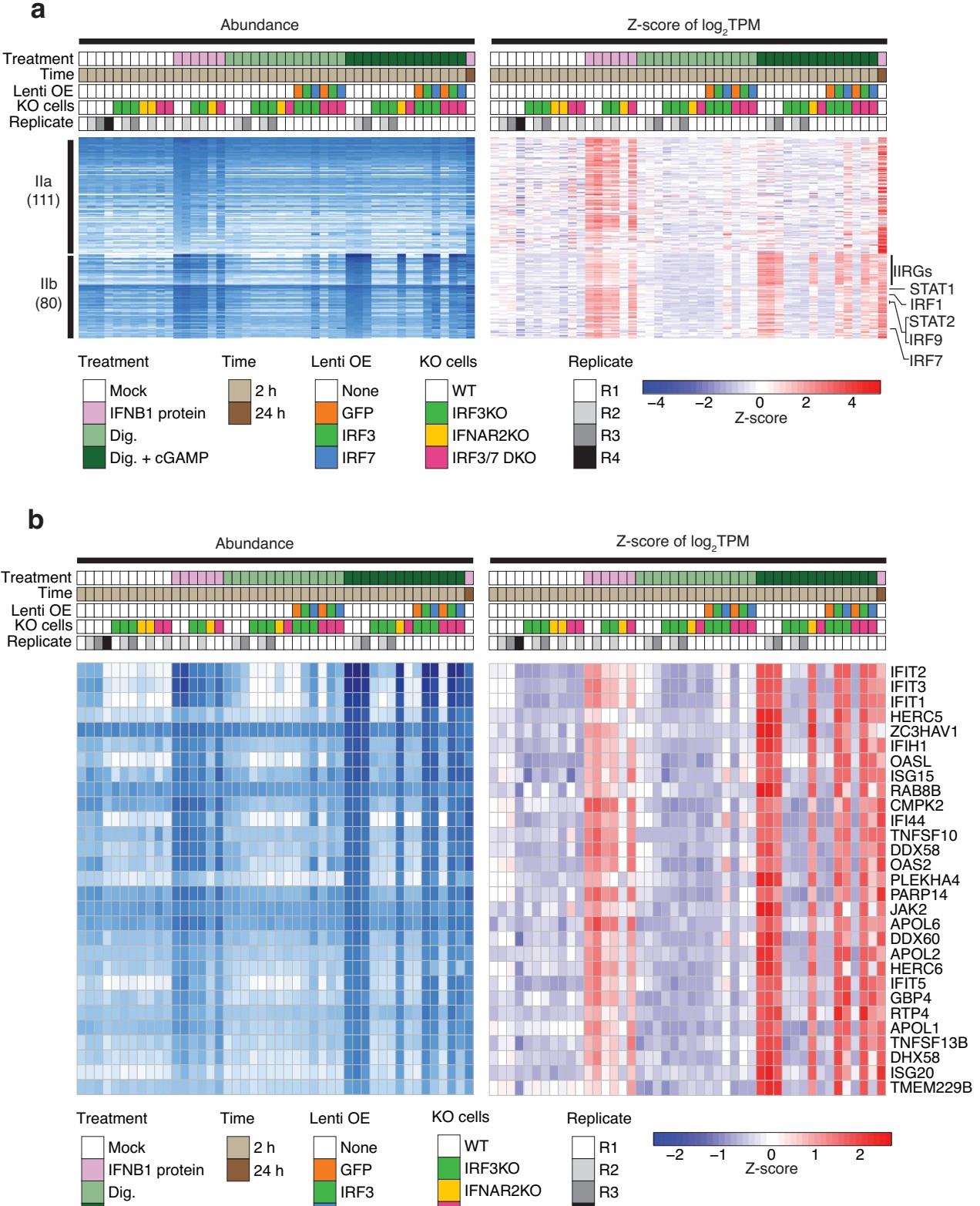

**Fig. 4 | Validation of Group II (IIa and IIb) ISGs using IFNAR2 KO, IRF KO and OE in THP1 monocytes. a** All Group II ISGs required IFNAR2 for induction by IFNB1 protein in the absence of cGAMP. Group IIb genes representing ISGs induced by IFNB1 protein or cGAMP in wild-type THP1 were impacted by IFNAR2 KO as well as IRF KO and OE. A subset of 29 Group IIb ISGs (IIRGs) also inducible by IRF3, retained strong induction (log$_2$FC > 2) in IFNAR2 KO sample following cGAMP treatment and are marked with a black side bar. **b** IIRGs are shown in expanded view. IRF3 or IRF7 OE in KO samples rescued their induction following cGAMP treatment.

distinguished by cell state and treatment conditions (Fig. 1c, Supplementary Fig. 4). Isogenic IRF and IFNAR2 KO cell lines clustered with wild-type THP1 cells unless the treatment response depended on the KO gene. For instance, cGAMP-treated IRF3 KO and IRF3/7 DKO samples clustered together with mock-digitonin-treated sample subgroup and IFNB1-protein-treated IFNAR2 KO sample clustered with the group of untreated samples. We therefore considered RNA profiles of isogenic lines that clustered into sample subgroups equivalent to biological replicates in subsequent differential gene expression (DGE) analysis.

The seven sample subgroups underwent group-wise DGE analyses to identify regulated genes with statistical significance (adjusted p-value < 0.05, Supplementary Data 3, Supplementary Fig. 5). This included comparisons between monocyte and macrophage cell states, and between cell-state-specific mock and treated conditions. We identified 3039 significantly regulated genes (≥2-fold) and average transcript abundance >20 TPM in the induced state (Fig. 1d). Monocyte-to-macrophage terminal differentiation resulted in 1108 up- and 758 downregulated genes, indicating extensive transcriptional reprogramming. IFNB1 protein, digitonin, cGAMP, or LPS treatment resulted in 1214 up- and 560 downregulated genes in monocytes, some of which were regulated by more than one treatment. LPS treatment of macrophages yielded 369 up- and 93 downregulated genes, of which 237 and 26 were also up- and downregulated, respectively, by LPS treatment of monocytes. Given the short 2 h time points and the rapid transcriptional activation driving innate immunity responses, we did not explore downregulated genes. Also, we did not pursue the small subset of genes uniquely regulated in LPS-treated macrophages.

The induced genes in monocytes were further subdivided by whether they were uniquely regulated by one or more than one of the treatments, resulting in seven groups of genes (Fig. 2, Supplementary Data 3). Unsupervised clustering of samples and averaged z-scores of these gene groups organized samples according to treatment, unless their treatment response was impacted by KO of IRF or IFNAR2 (Supplementary Fig. 6A). The outliers were cGAMP-treated IRF3 KO and IRF3/7 DKO, which clustered with mock-treated controls, and IFNB1-protein-treated IFNAR2 KO, which clustered with untreated samples. These gene groups represent IRF-, STAT-, stress-, and NF-κB-dependent gene regulation or a combination thereof.

### Group I genes induced by IRF3 and IRF7 following cGAMP treatment

DGE comparing cGAMP- versus mock-digitonin-treated sample subgroups (Fig. 1c) identified 402 genes that were upregulated ≥2-fold. A subset of 274 of these genes, referred to as Group I, was exclusively upregulated in cGAMP-treated samples and showed no regulation in IFNB1-protein- or LPS-treated versus control subgroups (Fig. 2). Group I genes included highly induced IFNB1 and IFNL1 (≥1000 TPM) as well as SEMA3D, LEPREL1, and SIX1 (≥100 TPM), a set of genes with potential roles in innate immune responses (Fig. 3).

IRF3 KO or IRF3/7 DKO monocytes did not respond to cGAMP (Fig. 3a). Stable ectopic overexpression (OE) of IRF3 in IRF3 KO or IRF3/7 DKO cells restored cGAMP-dependent induction of most Group I genes, indicating that these were direct targets of IRF3. The transcript abundance of IRF7 was 3-fold less compared to IRF3, and IRF7 protein abundance was too low (Supplementary Fig. 6B, C) to induce Group I genes by cGAMP in IRF3 KO monocytes. IRF7 OE in IRF3 KO or IRF3/7 DKO monocytes induced 7 (10%) out of 88 Group I genes (≥4-fold) by cGAMP, including IFNB1 or CXCL11, and IFNA14 and ENPP2, which were not rescued by IRF3 OE (Fig. 3b, Supplementary Fig. 7A, B). IFNB1 protein-pretreated cells exposed to cGAMP induced all Group I genes including IRF7-selective IFNA14 and ENPP2 (Supplementary Fig. 7C). IRF7 is an ISG and was 6-fold increased by pre-treatment of wild-type monocytes with IFNB1 protein. IFNB1 protein-pretreatment applied to IRF3 KO cells and followed by cGAMP treatment was insufficient to induce IFNA14 or ENPP2. The induced IRF7 mRNA abundance was 3-fold less compared to IRF7 OE in IRF3 KO cells. Together,

these experiments suggest that IRF3 and IRF7 act cooperatively in Group I gene induction, perhaps as phosphorylated IRF3-IRF7 heterodimers[58].

We also knocked out IRF1, IRF2, IRF5, IRF7, IRF8, IRF9, or IFNAR2. Except for IRF8, gene expression of untreated cell states was comparable to wild-type. IRF8 is a transcription factor implicated in myeloid lineage specification[59,60]. IRF8 was 6-fold more abundant compared to IRF3 in wild-type monocytes (Supplementary Fig. 6B). IRF8 KO resulted in 5-fold upregulation of the protease CTSG compared to wild-type, and it became one of the most abundant transcripts in monocytes. However, comparing wild-type THP1 monocytes to macrophages, we observed a simultaneous 10-fold and 8-fold reduction of CTSG and IRF8, respectively. Comparing IRF8 KO to wild-type macrophage, CTSG was still 10-fold increased, implying IRF8 as a repressor of CTSG in either cell state. With respect to 2 h cGAMP treatment, KO of neither IRF1, IRF2, IRF5, IRF7, IRF8, IRF9, or IFNAR2 impaired Group I gene induction (Supplementary Fig. 8). The mRNA abundance of IRF1 and IRF2 was comparable to IRF7, and therefore possibly too low to impact Group I expression, while IRF5, IRF9, and IFNAR2 were comparable to IRF3 (Supplementary Fig. 6B).

### Group II genes induced by IFNB1 protein treatment (ISGs) also include a subset of IRF3- and IRF7-driven genes induced by cGAMP

DGE comparing 2 h IFNB1-protein- versus untreated sample subgroups (Fig. 1c) identified 173 genes that were upregulated ≥2-fold. Increasing the extracellular IFNB1 protein concentration 100-fold did not further increase the number or level of ISGs after 2 h treatment versus control (Supplementary Fig. 9A). Furthermore, we identified an additional 64 genes upregulated ≥4-fold after 24 h IFNB1 protein treatment versus control, representing ISGs with delayed induction kinetics, e.g. IFI27 (Supplementary Fig. 5A). The combined 237 ISGs were then subdivided into groups depending on their induction by one or more treatment(s) (Fig. 2).

Group IIa comprises 111 ISGs uniquely induced in 2 h IFNB1-protein-treated cells, but still quiescent in 2 h cGAMP treatment (Fig. 2, Group IIa). Group IIb comprises 80 ISGs, which were also induced by 2 h cGAMP treatment. Group IIa and IIb required IFNAR2 for their induction by IFNB1 protein in absence of cGAMP (Fig. 4a).

Group IIb genes were either directly induced by phosphorylated IRF3 and IRF7 and/or represented the onset of autocrine IFNAR-dependent IFNB1 protein signaling. To differentiate between these possibilities, we treated IFNAR2 KO cells with cGAMP and continued to observe the induction of a subset of 29 Group IIb genes, including the antiviral ISGs IFIT1, IFIT2, IFIT3, OASL, and ISG15 (Fig. 4b). We refer to these dual-regulated genes as IRF3/7-IFNAR-response genes or IIRGs. The IIRG induction in TPM was also up to 10-fold higher by cGAMP compared to IFNB1 protein treatment. Induction by cGAMP of Group IIb genes was abolished by IRF3 KO or IRF3/7 DKO and rescued by IRF3 or IRF7 OE. The remaining Group IIb genes are IFNAR-dependent rapidly induced ISGs and include all key regulators of feed-forward amplification, such as IRF1, IRF7, IRF9, STAT1, and STAT2.

KO of IRF1, IRF2, IRF5, IRF7, or IRF8 did not alter Group II gene induction by IFNB1 protein or cGAMP treatment (Supplementary Fig. 9B). The impact of IRF9 KO will be described in a separate section below.

### Group III stress response genes observed upon digitonin permeabilization for cGAMP delivery

Digitonin used for cellular permeabilization to facilitate cGAMP entry into cells caused cellular stress and resulted in upregulation of 321 genes, 183 of which were uniquely upregulated by digitonin (Fig. 2, Group III). Group III genes included membrane-localized annexin ANXA1, heat shock protein HSP5A, and transcription factors FOSB, KLF2, and KLF6 (Supplementary Fig. 5B) and excluded interferons and ISGs (Supplementary Fig. 1D). None of the Group III genes were impacted by any IRF or IFNAR2 KO (Supplementary Fig. 10A).

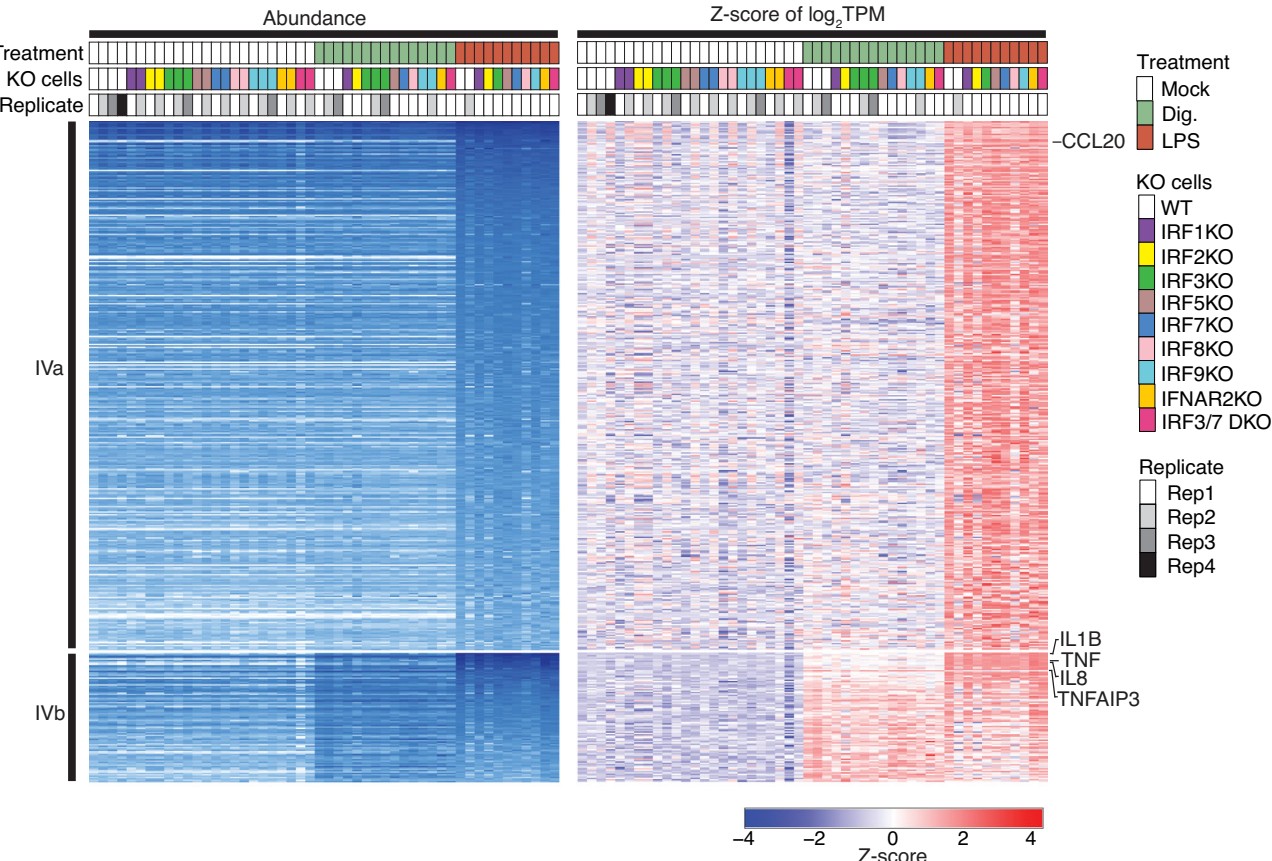

**Fig. 5 | Group IVa and IVb NF-κB-dependent genes are unaffected by IRF or IFNAR2 KOs in THP1 monocytes.** Group IVa genes exclusively induced by LPS and Group IVb genes primarily induced by LPS genes with minor co-induction by digitonin in wild-type and various KO THP1 monocytes. Induction of these genes remained unaffected by the KO of IRF or IFNAR2. Representative key pro-inflammatory genes are indicated by name.

## Group IV NF-κB-dependent genes induced by LPS

LPS treatment yielded 568 upregulated genes (Fig. 1c, Supplementary Fig. 5D), of which 389 were only induced by LPS treatment (Fig. 2, Group IVa) while another 95 genes were also induced albeit to a lesser magnitude by digitonin treatment (Fig. 2, Group IVb). Group IVa and IVb comprise many classical NF-κB-dependent pro-inflammatory genes such as *CCL4*, *CCL20*, *IL1B*, *IL8*, *TNF*, and *TNFAIP3*, reconfirming LPS as strong TLR4-mediated activator of the NF-κB pathway in THP1 monocytes[15]. The induction of Group IV genes was not compromised by KO of IRF or IFNAR2 genes (Fig. 5).

## Group V genes induced by cGAMP or LPS

We identified 9 genes (Fig. 2, Group V) that were induced by cGAMP or LPS and nominally by digitonin. These genes were well expressed covering a range of 400–5500 TPM, including the secreted cytokine *CCL3* and *CCL5* and transcriptional regulator *NFKBIZ* (Fig. 6a). Group V gene induction required IRF3 but not IFNAR2 in cGAMP-treated samples. OE of IRF3 in IRF3 KO cells rescued their induction, indicating that these genes are direct targets of IRF3. Furthermore, LPS-mediated upregulation of these genes was independent of IRF3 or IFNAR2, suggesting these genes are also direct targets of NF-κB-mediated signaling.

## Group VI genes induced by cGAMP, IFNB1 protein, or LPS

A group of 12 genes (Fig. 2, Group VI) was induced by treatment with either cGAMP, IFNB1 protein, or LPS, indicating promoter or enhancer elements targetable by multiple transcription factors (Fig. 6b). This included the cytokine *CXCL10* and ISGs not fitting into Group II, e.g. *GBP5*. IFNB1-protein-induced activation of *CXCL10* required IFNAR2, but not IRF3. The induction following cGAMP treatment required IRF3

but not IFNAR2. Reconstitution of IRF3 or IRF7 in IRF3 KO cells restored its induction, indicating *CXCL10* is also direct target of IRF3 and IRF7. Finally, LPS-induced activation of this gene was nominally affected by the absence of IRF3 or IFNAR2, also indicating its NF-κB pathway-dependent regulation.

## Group VII genes with atypical induction profiles

A group of 61 genes (Fig. 2, Group VII) could not be confidently assigned to a particular group because of overlapping induction by various treatments and/or partial response to IRF or IFNAR2 KOs. This included 34 ISGs (e.g. *CCL2*, *ICAM1*, *GBP2*) that required IFNAR2 for induction following IFNB1 protein treatment. However, these ISGs also showed IFNAR2-independent induction by digitonin and/or LPS.

## ISGs induced by IFNB1 in absence of IRF9

KO of IRF9 severely blunted the IFNB1 protein response of most ISGs (Group II, VI, and VII). IRF9 and phosphorylated STAT1 and STAT2 are required for the induction of the majority of ISGs[13]. 39 (22%) out of 173 ISGs induced by 2 h IFNB1 protein treatment were impacted less than 1.5-fold by IRF9 KO (Supplementary Fig. 10B, Supplementary Data 4), including IRF1 (Supplementary Figs. 2A, D). To assess if these IRF9-independent ISGs were also STAT-dependent, we reevaluated RNA-seq data from STAT1 and STAT2 KO in HeLa cells obtained 16 h after IFNB1 protein treatment[61]. We found only 5 out of these 39 genes were induced in IFNB1-protein-treated wild-type HeLa cells ($\log_2$FC > 1, pAdj < 0.05, RPM > 20). These genes were no longer induced by IFNB1 protein treatment in STAT1 KO HeLa cells, while STAT2 KO cells showed a reduced induction compared to wild-type cells, consistent with the idea of STAT1 homodimers being responsible for their expression.

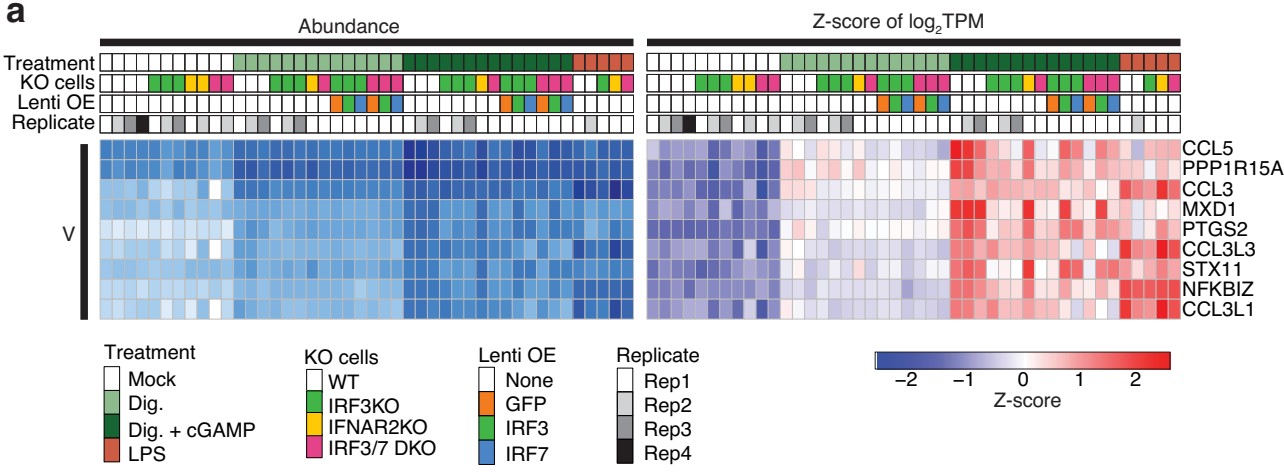

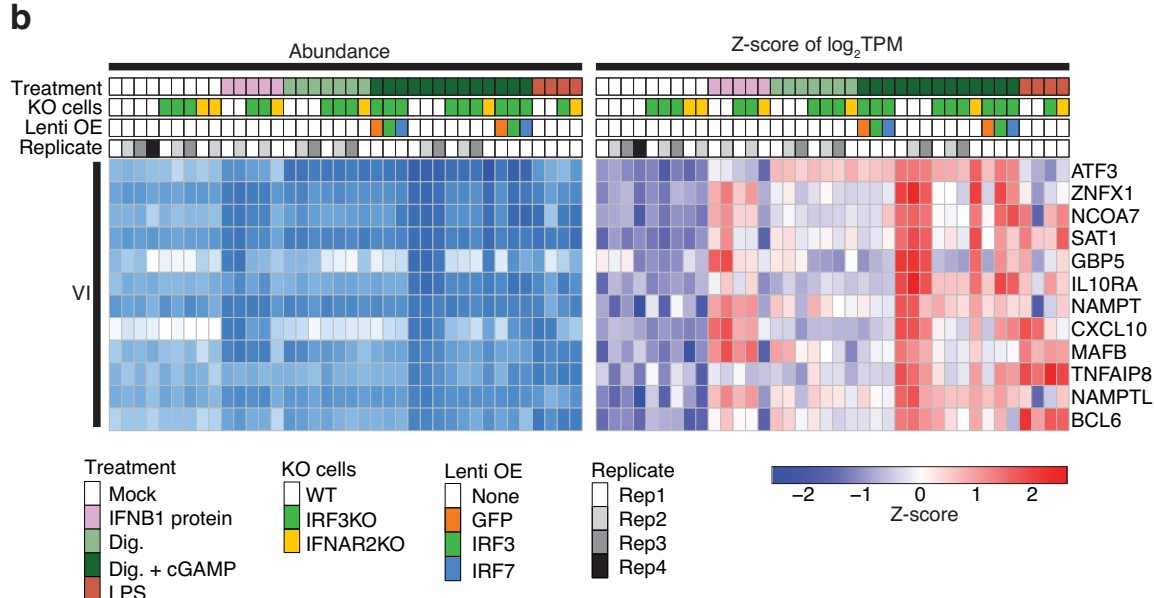

**Fig. 6 | Validation of Group V and VI co-induced genes using IFNAR2 KO, IRF KO and OE in THP1 monocytes. a** Group V genes co-induced by cGAMP or LPS in wild-type THP1 cells were impacted by IRF KO and OE. Their induction depended on IRF3 in cGAMP-treated samples but was IRF3-independent in LPS-treated samples. **b** Group VI genes co-induced by cGAMP, IFNB1 protein, or LPS in wild-type THP1 were impacted by IFNAR2 KO as well as IRF KO and OE. IFNAR2 was required for IFNB1-protein-induced activation but was dispensable in cGAMP-induced activation. IRF3 OE restored activation in response to cGAMP.

## Genes induced by LPS in PMA-differentiated macrophage THP1 cells

LPS treatment in macrophage THP1 cells yielded 369 upregulated genes (Fig. 1c), 283 of which were also captured by Group I to VII in monocyte THP1 cells (Fig. 7). These included 181 genes of the NF-κB response Group IV, and 29 of the cGAMP-activated Group I and IIb genes, such as *IFNB1*, *IFIT1*, *IFIT2*, and *IFIT3*, indicating an IRF3- and/or IRF7-mediated response along with the dominant LPS-driven NF-κB-mediated response. Co-induction of fast response Group IIb ISGs was weak, mirroring the 18-fold reduced *IFNB1* abundance in TPM compared to cGAMP-treated monocytes. The IRF3 KO macrophage THP1 cells were blunted for *IFNB1* expression and early ISG response.

## Discussion

Dissecting inflammatory gene expression has important implications for drug developers selectively targeting these pathways and thereby controlling specific gene sets for therapeutic intervention. Inhibiting innate-immune-signaling including interferon and cytokine expression will control different gene sets (Group I and IIRGs) compared to blocking interferon- or cytokine-receptor-signaling (Group IIa and IIb). Our approach unmasked early and direct innate immune transcriptional responses from subsequent overlapping autocrine and paracrine receptor-mediated responses.

Group I and Group II genes were induced 2 h post cGAMP or IFN protein stimulation, respectively, sharing a set of 80 genes induced by either treatment (Group IIb). Group I genes included the well-established type I and III IFN genes[62] requiring IRF3 or IRF7. By treating IFNAR2 KO cells with cGAMP, we identified 29 dual-responsive IIRGs within Group IIb, the remainder representing rapidly induced ISGs, including those with established feed-forward functions (*IRF1*, *IRF7*, *IRF9*, *STAT1*, and *STAT2*). The IIRGs include *IFIT1*, *IFIT2*, *IFIT3*, *ISG15*, *OAS2*, and *IFI44*, which prevent viral replication of infected cells[63]. While these genes were originally described as ISGs, their unexpected induction was previously noted upon expression of mutant and constitutively active IRF3 in human Jurkat T lymphocyte cells unable to produce IFNs[38]. IRF3-dependent but IFN-independent induction of *IFIT1*, *IFIT2*, *IFIT3*, and *ISG15* was also demonstrated in human foreskin fibroblast cells engineered to express

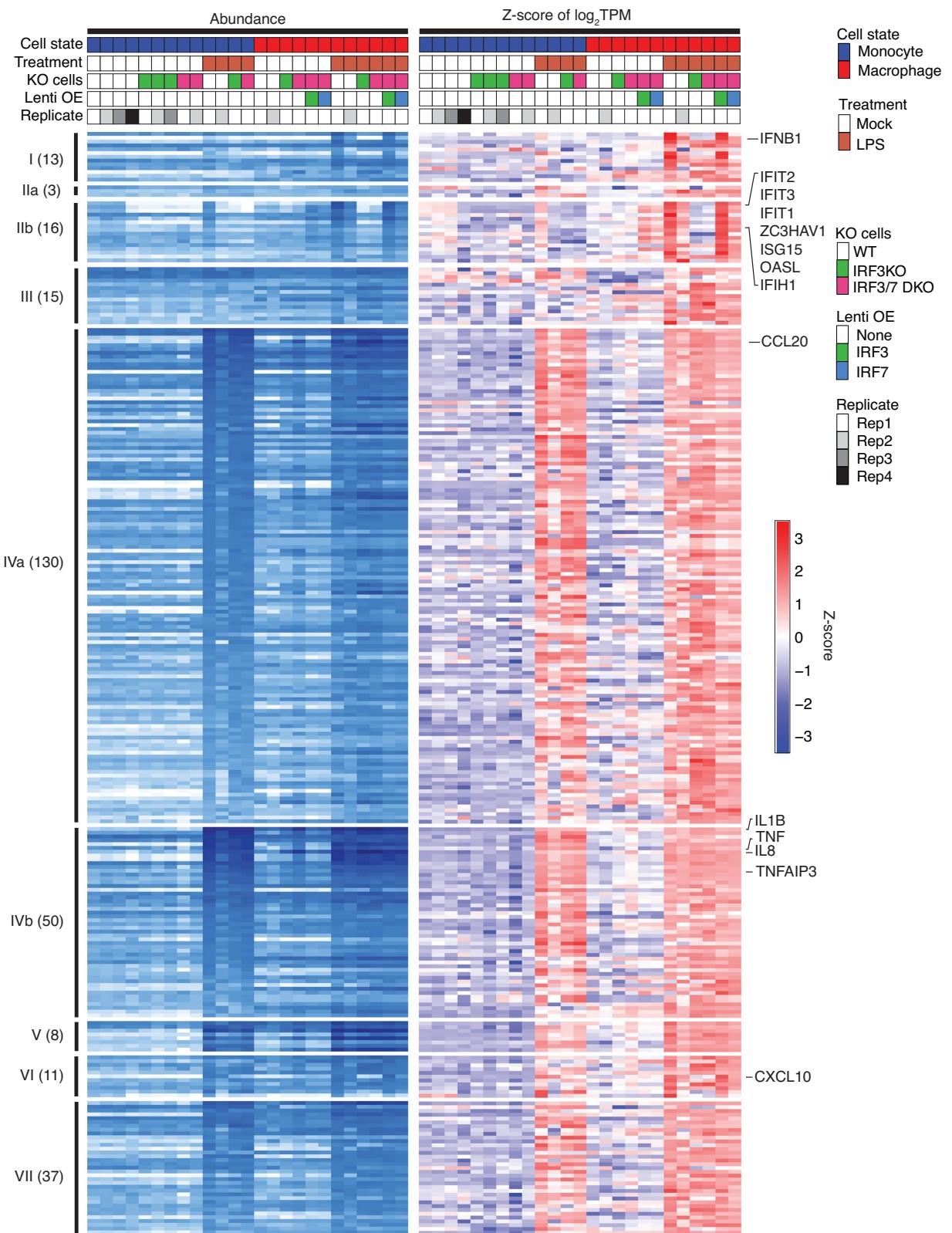

**Fig. 7 | LPS-induced TLR4 activation drives IFNB1 and ISG expression in THP1 macrophages.** 283 genes upregulated by LPS in THP1 macrophages showing overlapping induction by various treatments in THP1 monocytes (Fig. 2). Numbers in parentheses denote the total number of overlapping genes identified per group. Several Group I and IIb genes are also induced by LPS in THP1 macrophages, including IFNB1 and multiple ISGs; their induction is IRF3-dependent.

STAT1-degrading parainfluenza virus 5 V protein during human cytomegalovirus infection[64]. Importantly, we observed up to 10-fold stronger upregulation of these antiviral genes by cGAMP-mediated IRF3 or IRF7 innate immune activation than by IFNB1 protein exposure. While we used optimized conditions for recording innate immune responses, we note that Vesicular Stomatitis Virus (VSV) infection of THP1 cells yielded similar innate immune response (Group I and II) with respect to transcript abundances after 24 h, including IFNA gene induction[9].

This study also identified several other IRF-dependent Group I genes, which merit further functional interrogation. OTUD1 was recently characterized as a negative regulator of type I IFN induction by deubiquitinating IRF3[65]. The E3 ubiquitin ligases RNF149[66] and PELI1[67] may play similar roles in innate immune regulation. Other Group I genes are implicated in various non-immunomodulatory functions such as neural development (SEMA3D)[68], collagen assembly (LEPREL1)[69], and mammalian auditory system development (SIX1)[70].

KO of IRF1, IRF2, or IRF5 showed no impairment of cGAMP- or LPS-mediated IFN responses in THP1 monocyte or macrophage cells. Perhaps these IRFs require additional cofactors absent in THP1 cells or they play roles in other innate sensing pathways we did not study. IRF5 has been implicated in TLR7-activated IFN induction of THP1 cells[29], however, we found TLR7 not expressed (≤2 TPM). We were also unable to note a role of IRF8 in enhancing IFNB1 induction described for human blood monocytes[30] or impairing STING1 activation noted in THP1 cells[71]. Interestingly, untreated IRF8 KO cells showed highly elevated CTSG expression, a serine protease otherwise localized in azurophil granules. The CTSG protease abundance in IRF8 KO cells was so profound that it required additional protease inhibitor treatment to prevent protein degradation during cell lysate preparation. CTSG is also highly expressed in mast cells[72] and proliferating neutrophil progenitors[73], where IRF8 expression is absent or modest, respectively. In contrast, plasmacytoid and myeloid dendritic cells[74] show high IRF8 expression but do not express CTSG. Finally, we found that IFNB1 induction in THP1 macrophage cells following LPS-mediated activation of TLR4 pathway is dependent on IRF3 but not on IRF1[75].

A recent study reported that the NF-κB pathway is co-activated by engagement of the CGAS-STING1 in THP1 cells[76], based on RELA phosphorylation following 2 h treatment with a phosphorothioate-modified cyclic dinucleotide. Using unmodified cGAMP, we and others did not make this observation[77].

IRF3 or IRF7 KO in human epithelial HT1080 cell line enhanced NF-κB-dependent pro-inflammatory gene induction following LPS treatment[78,79]. Using THP1 monocyte or macrophage cells, our transcriptional responses upon LPS treatment were unaffected by IRF KO or DKO. These discrepancies may be attributed to differences in cellular systems.

IRF9-independent ISGs identified in this study may reflect IFNAR-mediated activation of cryptic DNA binding sites or alternative transcription factor complexes, such as phosphorylated STAT1 homodimers known to bind to GAS elements in the promoter region leading to induction of a subset of ISGs (ICAM1 and IRF1)[12,13,80]. Although primarily induced by IFNγ (IFNG), STAT1 homodimers have also been reported to be activated by type I IFNs. We noticed that most of our IRF9-independent ISGs were not induced in HeLa cells following IFNB1 protein treatment. This discrepancy may be attributed to cell-type-specific epigenetic differences and/or alternative transcription and enhancer complexes, or differences in observation time (2 h versus 16 h).

On a cautionary note, while CXCL10 is an established marker of the M1 pro-inflammatory state of macrophages[81], it is also widely employed as a surrogate marker of the activation of innate immune sensing pathways and IFN protein signaling[82–84]. Activation of a CXCL10 promoter-luciferase reporter construct by NF-κB or IRF3 during Hepatitis C virus infection had been reported previously[85]. We observed induction of CXCL10 by either LPS, cGAMP, or IFNB1 protein 2 h post treatment, however, the abundance or TPM of CXCL10 mRNA 2 h post treatments was about 100-fold less compared to 16 or 24 h exposure of THP1 monocytes to cGAMP[8], VSV[9] or a

6 h treatment with IFNγ-LPS of PMA-differentiated THP1 macrophages[86]. High level CXCL10 expression may implicate a form of feed-forward regulation established upon prolonged cGAMP treatment or viral infection partially engaging signaling factors otherwise fully activated upon IFNγ-LPS treatment driving the M0- to M1-transition of macrophages. CXCL10 regulation appears distinct from the IRF7-driven feed-forward loop considering its modest expression during otherwise high-level IFNA and IFNB1 expression in cGAMP-stimulated IRF7-OE THP1 monocytes.

The highly upregulated genes in this cell culture study correlated well with those induced in circulating leucocytes of viremic patients[87], though it missed induction of some cell-lineage-specific genes expressed by granulocytes/neutrophils (e.g. CXCR2, FCGR3B). Therefore, it might be beneficial expanding our approach to other immune cells and cell types in uncovering cell-specific innate immune transcriptional response.

In conclusion, innate immune stimulation followed by transcriptomic analysis using wild-type and isogenic gene KO cell lines identified distinct transcriptional response groups of genes characteristic of the underlying specific signaling pathways engaged upon viral, bacterial, or sterile (i.e. self) PAMP-triggered inflammation. Outlining these pathways and their unique as well as dual- or multiple-regulated gene groups will benefit inflammatory drug discovery. We have summarized these observations in Supplementary Data 5 for use in gene ontology and illustrated its application quantifying which innate immune pathways are active during viral replication in THP1 cells (Supplementary Fig. 11). Given that degenerative or fibrotic medical conditions have a chronic inflammatory component[88], inhibition of the innate immune response may be broadly beneficial to control disease progression.

## Materials and methods
### Antibodies and reagents
Proteins were detected by Western blotting using antibodies from Cell Signaling Technology, IRF1 8478S, IRF7 4920S, IRF8 5628S, IRF9 76684S, STAT1 9175S, pSTAT1-Y701 9167S, pIRF3-S386 37829S, pIRF3-S396 4947S, pTBK1-S172 5483S, pSTING-S366 19781S, NF-κB-RELA 8242S, NF-κB-pRELA-S536 3033S; R&D Systems, IRF2 AF4049; Abcam, IRF3 ab68481, IRF5 ab33478; Proteintech, GAPDH 60004-1-lg; Sigma-Aldrich, FLAG F3165; Dako, polyclonal goat anti-rabbit Immunoglobulins/HRP P0448, polyclonal goat anti-mouse Immunoglobulins/HRP P0447. Reagents were from InvivoGen, cGAMP vac-nacga23; PBL assay science, recombinant IFNB1 11415; Sigma-Aldrich, LPS L3129-10MG.

### Cell culture
THP1 cells from ATCC TIB-202 were cultured in Gibco RPMI 1640 supplemented with Sigma-Aldrich 10% v/v heat-inactivated fetal bovine serum (FBS), 100 U ml$^{-1}$ penicillin, 100 µg ml$^{-1}$ streptomycin, and 2 mM L-glutamine. HEK293T/17 cells were cultured in Thermo Fisher Scientific high glucose DMEM 11965118 supplemented with 10% v/v FBS, 100 U ml$^{-1}$ penicillin, 100 µg ml$^{-1}$ streptomycin and 2 mM L-glutamine.

### Cell lysis and western blot analysis
Cells were collected by centrifugation and washed with PBS to remove residual culture media. Cell pellets were incubated on ice for 15 min with 50 µl of lysis buffer per million cells. The lysis buffer contained 50 mM Tris-HCl pH 7.5, 150 mM NaCl, 2 mM EDTA, 2% NP40, 0.5 mM DTT, and was supplemented with 1 tablet of cOmplete EDTA-free protease inhibitor cocktail (Roche, 11836170001) and 1 tablet of PhosSTOP (Sigma Aldrich, 4906845001) per 50 ml. Lysis buffer for IRF8 KO cells was further supplemented with 2 mM PMSF and 4 mM AEBSF. Cell lysates were cleared by centrifugation at $13,000 \times g$ at 4 °C for 15 min. Supernatant protein concentration was measured using the Thermo Fisher Scientific BCA protein assay kit 23225. For Western blot analysis, 20–40 µg of total protein per sample was diluted in 1× SDS loading dye (1.7% SDS, 5% glycerol, 0.002% bromophenol blue, 60 mM Tris-HCl pH 6.8, 100 mM DTT), separated on a 12% SDS-polyacrylamide gel, and transferred to a nitrocellulose membrane by electroblotting using a Bio-Rad semi-dryer blotter. Membranes were

blocked with 5% BSA in 1× TBS-0.1% Tween for 3 h and incubated with the primary antibody overnight at 4 °C followed by three washes in 1× TBS-0.1% Tween. Secondary antibody incubation was performed using horseradish peroxidase-conjugated secondary antibody at room temperature for 3 h followed by three washes of 1× TBS-0.1% Tween. Finally, chemiluminescent detection was performed using a GE Amersham Imager 600 to visualize the blots.

### Generation of THP1 KO cell lines
CRISPR-Cas9-mediated genome editing of THP1 cells was performed as described[89] with few modifications. Briefly, 2.5 million THP1 cells were pelleted, washed once with PBS at ambient temperature and fresh pre-warmed culture media, and resuspended in 250 µl Gibco Opti-MEM 31985070. 2.5 µg pL-U6-gRNA-GFP and 2.5 µg pRZ-CMV-Cas9 plasmid were combined in a BIO-RAD 0.4 cm-gap sterile electroporation cuvette 1652081 together with the resuspended THP1 cells. Electroporation was performed using BIO-RAD Gene Pulser II using an exponential pulse at 250 V and 950 µF. Electroporated cells were gently resuspended and transferred into a 6-well-plate containing 3.75 ml of pre-warmed 50% THP1-culture-conditioned media supplemented with 20 µM human cGAS inhibitor InvivoGen G140 inh-g140. The inclusion of the cGAS inhibitor improved cell viability during and after transfection. After 24 h, live GFP$^+$/mCherry$^+$ cells were sorted using a BD FACS Aria device. Double-positive cells were sorted and individually placed into a 96-well-plate containing 70% conditioned media. Colonies were grown until visible to the naked eye, after which they were transferred into larger plates for expansion and subsequently tested for gene KO.

### Lentivirus production and lentiviral gene transduction
Four million HEK293T/17 cells were plated into a 10-cm cell culture plate. Lentiviral particle production was initiated 24 h after seeding by co-transfecting 5 µg Origene pLenti-EF1a-C-Myc-DDK-P2A-Puro empty plasmid PS100120 or pLenti-EF1a-C-Myc-DDK-P2A-Puro-GOI, along with 3.75 µg Addgene psPAX2 12260 and 1.25 µg pMD2.G 12259 packaging plasmids, using Thermo Fisher Scientific Lipofectamine 2000 11668019. The medium was replaced 24 h post-transfection. Viral-particle-containing supernatant was collected 48 and 72 h post-transfection, cleared by centrifugation at $500 \times g$ for 10 min at room temperature to remove cellular debris, and then passed through a Fisher Scientific 45 µm filter 09-720-005. The filtrate containing the viral particles was complemented with 1/4 volume of ice-cold System Biosciences PEG-it LV810A and incubated for 24 h at 4 °C. The viral particles were pelleted by centrifugation at $1500 \times g$ for 30 min at 4 °C. The supernatant was aspirated, and the viral pellet resuspended in 500 µl of PBS.

Lentiviral transduction was performed by spinfection in a 2 ml final volume in 12-well cell culture plates containing 1 million THP1 cells, 40 µl of virus, and 10 µg ml$^{-1}$ polybrene at $1258 \times g$ for 90 min at 37 °C. 24 h after infection, cells were transferred to a 6-well plate containing 2 ml of fresh culture media and incubated for an additional 24 h. Cells were pelleted, resuspended in 3 ml fresh media, and selected with 2 µg ml$^{-1}$ puromycin. Expression of FLAG-tagged proteins was confirmed by Western blot using gene- or FLAG-specific antibodies.

### Plasmids
gRNA plasmids for CRISPR KOs were built following established protocols[89,90]. The gRNA sequence targeting the coding region of the gene of interest was designed using online tool (http://crispor.tefor.net)[91]. The gRNA, with the PAM sequence flanked by constant sequences (5′GGAAAGGACGAAACACCG-gRNA-PAM-TTTAGAGCTAGAAA-TAGC-3′), was inserted into pL-U6-gRNA-GFP using the ligation-independent assembly method[90].

For generation of lentiviral overexpression plasmids, the corresponding coding sequence was amplified from cDNA prepared from 20 h IFNB1-protein-treated THP1 cells, introducing AscI/MluI restriction sites for GFP, IRF3, and IRF9, or AscI/NotI restriction sites for IRF7. The GFP PCR

product was amplified from Addgene pcDNA5_FRT-GFP 127108. The resulting PCR product and Origene pLenti-EF1a-C-Myc-DDK-P2A-Puro PS100120 were digested with the respective restriction enzymes. Digested plasmids were gel purified and ligated into pLenti-EF1a-C-Myc-DDK-P2A-Puro, followed by sequence verification. All plasmids created or used in this study are available from Addgene.

### RNA isolation
Total RNA for RT-qPCR and RNA-seq analysis was isolated from each sample using TRIzol (Thermo Fisher) following manufacturer's protocol. Briefly, cells harvested after various treatments were pelleted, washed with PBS, and RNA was extracted using TRIzol (Thermo Fisher).

### RT-qPCR gene expression analysis
RNA isolated from THP1 cells was quantified using RT-qPCR. Briefly, 800 ng RNA was reverse-transcribed into cDNA in a 20 µl final reaction volume using 2.5 µM oligo(dT)$_{20}$ primer and 10 U µl$^{-1}$ Superscript III (Thermo Fisher) for 50 min at 50 °C. Quantitative PCR was performed on an Mx300P qPCR System (Agilent Technologies) using 1/80th of the reverse transcribed material as input for qPCR. Expression of mRNAs was measured in triplicate per sample. Threshold cycle ($C_T$) values for selected mRNAs were normalized to TUBA1B $C_T$ values and used to calculate $\Delta C_T$. Relative mRNA expression levels were calculated using the $\Delta\Delta C_T$ method ($2^{\Delta\Delta C_T}$) and shown relative to digitonin only control (Supplementary Figs. 1D, 7A, B) or media (Supplementary Fig. 1E, F, H). Primer sequences are provided in Supplementary Table 1.

### Sequencing and analysis
Oligo(dT)-selected RNA was converted into cDNA using the Illumina Truseq RNA Sample Preparation Kit v2. Sequencing was performed at The Rockefeller University Genomics Resource Center on the Illumina NextSeq 500 platform with 2 × 75 nt paired-end.

Annotation was performed using an in-house custom pipeline. Sequencing file quality was assessed with FastQC software (http://www.bioinformatics.babraham.ac.uk/projects/fastqc). Quality-filtered reads were mapped using BOWTIE[92] allowing a maximum of two mismatches, against an in-house curated transcript database of the human transcriptome, genome (hg.38). A hierarchical approach was used to assign each read in the order of mRNA, lincRNA, tRNA, miRNA, rRNA, and uncharacterized genomic sequences. Read counts for the mRNA and lincRNA categories were selected for downstream differential gene expression analysis using DESeq2 or normalized to TPM. Data representation and visualization were performed using the R packages ggplot2 and pheatmap.

### Pairwise expression analysis
Pairwise analysis was performed between the following samples: i) Mock-treated KO cell versus mock parental cells in both monocytes and PMA-differentiated cells, and ii) treated versus mock conditions for parental, KO or lentiviral GFP-, IRF3-, or IRF7-overexpressing cells within each treatment group. A pseudo count of 20 was uniformly applied for fold change (FC) calculations. A union of regulated genes (|FC|> 8 for monocyte-to-macrophage comparisons and |FC|> 4 for all others) was generated in each group and combined together for all comparisons. The final gene list for unsupervised clustering analysis included genes with TPM > 300 in at least one sample, resulting in 360 highly regulated genes. The selected 360 genes used for unsupervised clustering analysis are listed in Supplementary Data 3.

### Assessment of IRF dependence in gene groups
Averaged z-scores were used to assess the contribution of individual IRFs to the induction of each gene group. For each sample, z-scores were averaged across seven gene groups (Fig. 2) and subjected to unsupervised clustering to determine dependence of IRF within each group. Most KO samples exhibited averaged z-scores comparable to wild-type sample under the same treatment conditions (Supplementary Fig. 6A).

ISGs impacted by the absence of IRF9 were identified calculating the ratio of IRF9 KO to wild-type for IFNB1-protein-treated samples. Those with a ratio >0.7 representing retention of 70% or greater in induction were classified as IRF9-independent.

Dependence of IRF3 or IRF7 on group I genes (Fig. 3) was identified through pairwise FC analysis comparing cGAMP-treated IRF OE cells to digitonin-treated parental cells. The threshold selected for the classification of group I genes uniquely or commonly upregulated by IRF3 or IRF7 are indicated in Fig. 3 legend. A uniform pseudo counts of 1 was applied across all samples. The corresponding $\log_2$FC values are also presented in Supplementary Data 3.

## Statistics and reproducibility

RNA-seq replicates were not generated for all the samples in this study due to the extensive scale of the study encompassing numerous KO cell lines and treatment conditions. Samples with available replicates are indicated in Fig. 1B. Samples showing concordant RNA profiles of isogenic lines that were indistinguishable from co-clustered subgroup samples were considered as replicates for DGE analysis. All RT-qPCR data are expressed as mean of two or three technical replicates.

## Reporting summary

Further information on research design is available in the Nature Portfolio Reporting Summary linked to this article.

## Data availability

RNA-seq data are deposited at NCBI Short-Read Archive (SRA) under the BioProject number PRJNA1244336. The following public RNA-seq data series were also used in this study: GSE133139, GSE154596, GSE79272, GSE69906, GSE114966, GSE157052, GSE199674, GSE128113, GSE176261 and GSE130011. Expression values for all samples generated, analyzed, and included in this study (Supplementary Data 1-5), and numerical source data for the RT-qPCR plots (Supplementary Data 6), are publicly available in the Dryad repository [https://doi.org/10.5061/dryad.v15dv428j]. Uncropped blot images are provided in Supplementary Information as Supplementary Figs. 12–17.

## Code availability

No custom computer codes were generated in this study.

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

## Acknowledgements

T.T. acknowledges funding support from NIH AI141507. We also thank the Rockefeller University Genomics Resource Center and Flow Cytometry Resource Center.

## Author contributions

L.L. undertook the studies, analyzed the data, and prepared the manuscript under the supervision of T.T. P.M. performed RNA-seq data alignment and assisted in data analysis. A.G. assisted in RNA-seq data analysis. T.T. provided overall supervision of the study.

## Competing interests

The authors declare the following competing interests: T.T. is a cofounder and scientific advisor of Ventus Therapeutics. All other authors declare no competing interest.
