## [Transparent Peer Review file · Communications Biology]

Dissection of innate-immune-ligand- and interferon-protein-mediated transcriptional responses in human THP1 cell states

Corresponding Author: Dr Thomas Tuschl

Version 0:

Reviewer comments:

Reviewer #1

(Remarks to the Author)

This manuscript reports a comprehensive dataset of RNA abundance changes in response to innate immune stimulation of human THP1 cells. An extensive use of gene knockouts and re-expression validations for relevant transcription factors and receptors allows classification of potential response pathways for the identified gene induction patterns. The overall conclusions based on these data are largely consistent with what has been surmised by previous smaller analyses, although there are the inevitable individual inconsistencies with other studies, possibly due to cell state differences or differences in exact treatment protocols. As such, the datasets will add a valuable resource for further analysis of these pathways, due to the more comprehensive nature of the experiments and/or the use of more accurate and contemporary methods. As such, these data should garner wide interest in the field.

There are some areas where additional attention could increase the utility of the data.

1. An overarching question is the biological significance of the documented gene expression changes. For instance, the authors identify a set of genes activated directly by IRF3 in response to cGAMP as well as presumably by STATs in response to IFN β . It would be helpful to know to what extent there exists an IRF3-dependent but STAT-independent antiviral or inflammatory response and whether these physiological responses result in distinct biology. Given the overlapping feedback and feedforward pathways that regulate the overall antiviral response, the biological significance of gene expression changes at individual timepoints is difficult to fully appreciate. It is noted that robust IFN β gene induction was observed at 1h post stimulation and that robust STAT1 phosphorylation was seen by 2.5h; it is likely that feedforward pathways are already influencing gene expression at the 2h timepoint used for the majority of this study.
2. Perhaps related to the above comment, it is somewhat cumbersome to appreciate the magnitude of the gene expression changes due to the nature of the presentation, without delving into the supplemental tables. Perhaps the figures could be modified to be more helpful for the reader to understand example genes falling into each category and their degree of abundance.
3. It is well established that cGAMP can activate a NF- κ B response. It would be helpful to identify the reasons that this response is not observed in the present study of THP1 cells.
4. It is also well established that LPS stimulation of normal macrophages leads to the rapid induction of IFN β secretion. That IFNAR2-dependent induction of ISGs was not observed in this study following LPS stimulation deserves comment.
5. The pathway underlying IFN β -induced genes that are IRF9 independent remains unidentified. Is this due to STAT1 homodimers?
6. The authors hypothesize that IFN β induces only a mild antiviral state that prepares cells for a more violent IRF3- and IRF7-driven innate response and increased IFN gene expression. This is basically a restatement of the feedforward hypothesis where it has been shown that various ISGs (e.g., IRF7, STAT1, STAT2, IRF9, etc) are induced by IFN and render cells more responsive to subsequent stimulation by either virus (due to more IFN production) or IFN (due to more robust gene induction). What remains unclear is whether the direct activation of antiviral ISGs by IRF3 triggers a physiologically relevant antiviral state in absence of subsequent IFN production and response.

Reviewer #2

(Remarks to the Author)

Lama et al. present a comprehensive RNA-seq study aimed at elucidating the roles of IRF family members (IRF1, IRF2, IRF3, IRF5, IRF7, IRF8, IRF9) in transcriptional responses to cGAMP, LPS, and IFN- β in the human THP-1 cell line. The authors employed CRISPR/Cas9-mediated knockouts and lentiviral overexpression experiments to validate the functional rescue of individual IRF knockouts. In total, they generated an impressive dataset comprising 114 RNA-seq samples. Through this approach, the authors identified seven distinct transcriptional groups based on their unique or overlapping induction patterns in response to the different treatments. Within these groups, the dependence on specific IRF family members was systematically characterized. Surprisingly, the study revealed that four out of the seven IRF family members were largely dispensable for the transcriptional responses to the treatments analyzed.

The study is thoroughly performed, and the data are presented in a clear and accessible manner. While primarily descriptive, this work provides a valuable resource for understanding the transcriptional functions of IRF family members at an early time point following stimulation with cGAMP, LPS, or IFN- β .

Major comments:

- It appears that not all samples were performed in triplicate, and some samples may have been analyzed only once. If this is indeed the case, the authors should explicitly acknowledge this limitation in the manuscript.
- The criteria used to define "IRF9-dependent" (or dependence on any other IRF) are unclear. Did the authors apply a binary classification (e.g., induced vs. not induced), or did they also take into account quantitative differences, such as reduced fold induction in knockout (KO) cells compared to wild-type (WT) cells, or lower expression levels in treated KO versus WT cells? Clarification on this point would be helpful.

Minor comments:

- Supplementary Figure 6a: The X-axis labeling is incomplete in some of the graphs. Additionally, it is unclear why all digitonin-treated cells exhibit high expression levels of the analyzed genes (e.g., *Irfnb*). For improved clarity, relative expression levels (e.g., fold change) should be presented instead of Δ CT values for all RT-qPCR experiments.
- Table 1 (mentioned in line 113), Table 2 (mentioned in lines 153/154), and Table 3 (mentioned in line 162): These tables were not explicitly labeled, making it difficult to locate the ones referenced by the authors. I assume these Excel tables will be included in the supplementary materials, and they should be labeled accordingly. Furthermore, it would be beneficial to provide detailed descriptions and legends for all supplementary tables to enhance their clarity and usability.
- Line 210: The authors compare their data to a previous study using IRF9-KO THP-1 cells and report that, out of the 174 ISGs identified in the earlier study as IRF9-dependent, 62 ISGs were detected (threshold: TPM > 2) in the current study, with 41 of them also confirmed as IRF9-dependent. To support this comparison, the corresponding gene lists should be included in the supplementary materials.

Version 1:

Reviewer comments:

Reviewer #1

(Remarks to the Author)

This revised manuscript addresses the criticisms and suggestions from the prior review and presents a dataset and discussion that will be valuable for the scientific community. My only remaining suggestion relates to how the authors describe the relative importance of IRF3/7 direct gene induction versus IFN-stimulated (presumably STAT dependent) responses. The authors conclude "that an IFNB1 protein response is only leading into a "mild antiviral states" from a perspective of induction of direct acting antiviral genes, however, the concomitant upregulation of IRF7, IRF9 and other ISGs, including those involved in class I and class II antigen presentation, prepares cells for a possible infection and enables a stronger IRF7- and IRF9-assisted intracellular innate response and increased IFN gene expression." This statement is based on quantitative comparison of cGAMP v. IFN stimulated gene expression levels, but to conclude that physiologically relevant antiviral responses are primarily IRF3/7 driven after a largely priming role for IFN seems premature based on these experimental data. While there are well-documented feedforward pathways that regulate innate antiviral responses, to draw conclusions on the relative importance would require experiments in a more physiological setting, for instance, during a viral infection. Transfecting cells with cGAMP could bias the gene expression patterns that would be observed in a more biological context.

Reviewer #2

(Remarks to the Author)

The authors have thoroughly and sufficiently addressed all comments and concerns.

Version 2:

Reviewer comments:

Reviewer #1

(Remarks to the Author)

The text clarifications and new data address my former concerns.

Point-by-point referee response

Reviewer #1 (Remarks to the Author):

This manuscript reports a comprehensive dataset of RNA abundance changes in response to innate immune stimulation of human THP1 cells. An extensive use of gene knockouts and re-expression validations for relevant transcription factors and receptors allows classification of potential response pathways for the identified gene induction patterns. The overall conclusions based on these data are largely consistent with what has been surmised by previous smaller analyses, although there are the inevitable individual inconsistencies with other studies, possibly due to cell state differences or differences in exact treatment protocols. As such, the datasets will add a valuable resource for further analysis of these pathways, due to the more comprehensive nature of the experiments and/or the use of more accurate and contemporary methods. As such, these data should garner wide interest in the field.

We thank this reviewer for the thoughtful assessment of our manuscript and acknowledging the significance of new datasets in further advancing innate immune response research.

There are some areas where additional attention could increase the utility of the data.

*1. An overarching question is the biological significance of the documented gene expression changes. For instance, the authors identify a set of genes activated directly by IRF3 in response to cGAMP as well as presumably by STATs in response to IFN β . It would be helpful to **know to what extent there exists an IRF3-dependent but STAT-independent antiviral or inflammatory response** and whether these physiological responses result in distinct biology. Given the overlapping feedback and feedforward pathways that regulate the overall antiviral response, the biological significance of gene expression changes at individual timepoints is difficult to fully appreciate.*

We elaborated further on the question of biological relevance of IRF3- versus IFNAR-driven gene regulation and revised a paragraph to highlight in as much the innate IRF3 versus IFNAR pathway was able to induce antiviral acting genes. “Group IIb comprises 80 ISGs, which were also induced by 2 h cGAMP treatment (Fig. 2). These genes were either directly induced by phosphorylated IRF3 and IRF7 and/or represented the onset of autocrine IFNAR-dependent IFNB1 signaling (Fig. 5A). A subset of 29 Group IIb genes, including the antiviral ISGs IFIT1, IFIT2, IFIT3, OASL, and ISG15, continued to be induced in IFNAR2 KO cells upon cGAMP treatment (Fig. 5B). Their expression **in TPM** was also up to 10-fold higher by cGAMP compared to IFNB1 protein treatment. IRF3 KO or IRF3/IRF7 DKO cells did not induce Group IIb by cGAMP, and IRF3 or IRF7 OE rescued their expression. Therefore, these 29 Group IIb ISGs also represent IRF3- or IRF7-inducible genes. The remaining Group IIb genes are merely IFNAR-dependent rapidly induced ISG.”

*It is noted that robust IFN β gene induction was observed at 1h post stimulation and that robust STAT1 phosphorylation was seen by 2.5h; it is likely that **feedforward pathways** are already influencing gene expression at the 2h timepoint used for the majority of this study.*

We have updated our manuscript to justify the minimal influence by feed-forward pathways in the selected 2 h time point. We have revised and expanded the following paragraph: “A time course of cGAMP treatment using digitonin-permeabilized THP1 monocytes revealed a biphasic posttranslational response (Supplementary Figure 1B). The first phase of phosphorylation of STING1, TBK1, and IRF3 S386 was detectable at 0.5 h, plateaued at 2.5 h, and persisted up to 4.5 h post treatment. The second phase, marked by phosphorylation of STAT1 Y701, emerged 2.5 h post treatment. By contrast, direct IFNB1 protein treatment of THP1 cells showed a less than 30 min onset of STAT1 Y701 phosphorylation and lasted for about 4 h (Supplementary Figure 1C). The first phase following cGAMP exposure corresponded to the innate immune sensing as exemplified by IRF3-dependent IFNB1 protein synthesis, the second phase was caused by IFNAR signaling due to IFNB1 protein binding. The 2-h delay in STAT1 Y701 phosphorylation by cGAMP treatment versus direct IFNB1 protein exposure represents the time needed for IFNB1 gene transcription and translation. Similarly, increased protein levels of ISGs, such as IRF9, were only noticeable 3 h after IFNB1 protein treatment, and further increased and lasted for at least 8 h.”

2. Perhaps related to the above comment, it is somewhat cumbersome to appreciate the magnitude of the gene expression changes due to the nature of the presentation, without delving into the supplemental tables. Perhaps the figures could be modified to be more helpful for the reader to understand example genes falling into each category and their degree of abundance.

We employed both absolute (TPM) and relative abundance (z-score) representations across all figures to highlight the limitations of relying solely on relative expression, which can be misleading sometimes for genes with no basal expression. We appreciate the reviewer’s suggestion to improve figure clarity for readers. We have now updated Fig. 2 accordingly and included graphic representations of TPM values for some example genes (Fig. 2B) to better illustrate the degree of abundance in the revised version.

3. It is well established that cGAMP can activate a NF-κB response. It would be helpful to identify the reasons that this response is not observed in the present study of THP1 cells.

We disagree with the statement that it is well-established that cGAMP also activates NF-κB responses. There is a small set of genes that cGAMP as well as LPS can activate (group V genes) and at a first glance may be misinterpreted as activation of more than one pathway. However, prototypical NF-κB-mediated genes uniquely induced by LPS in group IV (IL1B, CXCL8/IL8, TNF, TNFAIP3) are not induced by cGAMP. Reporter assays relying on STING overexpression indicate that NF-κB promoter activity was approximately 100-fold lower than IRF3-responsive promoters and 80-fold lower than ISRE promoter¹ (Figure screenshot shown below), indicating minimal if any NF-κB activation. Another study also noted the absence of cGAMP-induced NF-κB activation in human THP1 cells, murine RAW264.7 cells, and BMDM cells² (Figure screenshot shown below) after 4 h of treatment, consistent with our experimental conditions. These findings suggest the absence of cGAMP-induced NF-κB activation in THP1 cells. We have revised our manuscript to address these earlier findings and the reviewer’s concern.

Figure 2 from Ishikawa et al¹.

Figure 3C from Zhang et al².

4. It is also well established that LPS stimulation of normal macrophages leads to the rapid induction of IFN β secretion. That IFNAR2-dependent induction of ISGs was not observed in this study following LPS stimulation deserves comment.

Although we noticed robust induction of IFNB1 gene (~100 TPM) following LPS treatment in THP1 macrophages, it was still 18-fold less than the amount induced by cGAMP in monocytes. Consequently, we also observed a markedly lower number of ISGs induced in the macrophages upon LPS stimulation compared to monocytes treated with cGAMP. We have incorporated this observation in the revised version.

5. The pathway underlying IFN β -induced genes that are IRF9 independent remains unidentified. Is this due to STAT1 homodimers?

We have commented on potential STAT1 homodimerization-dependent ISG induction in the revised manuscript.

6. The authors hypothesize that IFN β induces only a mild antiviral state that prepares cells for a more violent IRF3- and IRF7-driven innate response and increased IFN gene expression. This is basically a restatement of the feedforward hypothesis where it has been shown that various ISGs (e.g., IRF7, STAT1, STAT2, IRF9, etc) are induced by IFN and render cells more responsive to subsequent stimulation by either virus (due to more IFN production) or IFN (due to more robust gene induction). What remains unclear is whether the direct activation of antiviral ISGs by IRF3 triggers a physiologically relevant antiviral state in absence of subsequent IFN production and response.

We note for the reviewer that rapid induction of individual ISGs by IRF3 without IFN signaling has been reported by several studies. These papers are cited, but we report the entire IRF3-induced set of genes and its overlap with ISGs as we used RNA-seq analysis.

Reviewer #2 (Remarks to the Author):

Lama et al. present a comprehensive RNA-seq study aimed at elucidating the roles of IRF family members (IRF1, IRF2, IRF3, IRF5, IRF7, IRF8, IRF9) in transcriptional responses to cGAMP, LPS, and IFN- β in the human THP-1 cell line. The authors employed CRISPR/Cas9-mediated

knockouts and lentiviral overexpression experiments to validate the functional rescue of individual IRF knockouts. In total, they generated an impressive dataset comprising 114 RNA-seq samples.

Through this approach, the authors identified seven distinct transcriptional groups based on their unique or overlapping induction patterns in response to the different treatments. Within these groups, the dependence on specific IRF family members was systematically characterized. Surprisingly, the study revealed that four out of the seven IRF family members were largely dispensable for the transcriptional responses to the treatments analyzed.

The study is thoroughly performed, and the data are presented in a clear and accessible manner. While primarily descriptive, this work provides a valuable resource for understanding the transcriptional functions of IRF family members at an early time point following stimulation with cGAMP, LPS, or IFN- β .

We also thank this reviewer for the helpful comments.

Major comments:

- It appears that not all samples were performed in triplicate, and some samples may have been analyzed only once. If this is indeed the case, the authors should explicitly acknowledge this limitation in the manuscript.*

RNA-seq replicates were not included for all samples in this study due to the extensive scale of the work comprising numerous KO cell lines and treatment conditions. However, we included replicates for the most critical conditions and observations that directly support the conclusions of the study. We have explicitly acknowledged this limitation in the revised manuscript under “Statistics and Reproducibility” in the Methods section.

Importantly, we observed no significant changes in gene expression for most of the KO cell lines compared to the wild-type cells across various treatment groups (Supplementary Figure 4). Consequently, such isogenic KO cell lines indistinguishable in expression from wild-type cells can be considered equivalent to biological replicates. We have updated the manuscript to highlight this observation and to clarify the rationale for using KO cell lines as replicates.

- The criteria used to define "IRF9-dependent" (or dependence on any other IRF) are unclear. Did the authors apply a binary classification (e.g., induced vs. not induced), or did they also take into account quantitative differences, such as reduced fold induction in knockout (KO) cells compared to wild-type (WT) cells, or lower expression levels in treated KO versus WT cells? Clarification on this point would be helpful.*

We have clarified the criteria used to determine IRF dependence by adding a separate “Assessment of IRF dependence in gene groups” subsection under Methods in the revised version, as detailed below.

IRF9-dependence was assessed by comparing the reduced expression of ISGs in IRF9 KO cells relative to the wild-type parental cells following 2 h IFN β 1 protein treatment.

IRF-dependence across various gene groups was determined using two approaches:

- 1) Unsupervised clustering analysis of 360 most regulated genes across sample subgroups (Supplementary Figure 4), which consistently showed that most KO samples were indistinguishable from wild-type cells within the same treatment group.
- 2) Unsupervised clustering of averaged z-scores for seven gene groups – identified based on different treatments in monocytes comprising 1,214 genes across all monocyte samples (Supplementary Figure 6A), which yielded similar results, with most IRF KO samples clustering closely with wild-type samples. We included this new analysis and figure in the supplement.

Minor comments:

• *Supplementary Figure 6a: The X-axis labeling is incomplete in some of the graphs. Additionally, it is unclear why all digitonin-treated cells exhibit high expression levels of the analyzed genes (e.g., Ifnb). For improved clarity, relative expression levels (e.g., fold change) should be presented instead of ΔCT values for all RT-qPCR experiments.*

We have corrected the incomplete labeling for Supplementary Figure 6A (now titled Supplementary Figure 7A). Following the reviewer's guidance, we have also updated all RT-qPCR experiments and presented the results as "Fold change" for improved clarity in the revised version.

• *Table 1 (mentioned in line 113), Table 2 (mentioned in lines 153/154), and Table 3 (mentioned in line 162): These tables were not explicitly labeled, making it difficult to locate the ones referenced by the authors. I assume these Excel tables will be included in the supplementary materials, and they should be labeled accordingly. Furthermore, it would be beneficial to provide detailed descriptions and legends for all supplementary tables to enhance their clarity and usability.*

These tables (now titled Supplementary Data 1, 2, and 3) are now accessible to reviewers via the following private link in Dryad:

<http://datadryad.org/share/CzsuCpLEF90wIwDUtpxyrK4ivRWY2OVtvFwK8juRk9Q>

This link will be replaced with a permanent DOI upon publication.

• *Line 210: The authors compare their data to a previous study using IRF9-KO THP-1 cells and report that, out of the 174 ISGs identified in the earlier study as IRF9-dependent, 62 ISGs were detected (threshold: TPM > 2) in the current study, with 41 of them also confirmed as IRF9-dependent. To support this comparison, the corresponding gene lists should be included in the supplementary materials.*

Revisiting of the raw RNA-seq data of the previously reported study and analysis using our platform revealed an unexpected and not discussed strong and variable NF- κ B response across the wild-type and IRF9 KO macrophage datasets (6,000-18,000 TPM for IL1B) prior to IFNB1 protein treatment. Their reported IRF9-dependent ISG list was also devoid of several prominent ISGs (e.g. IFIT1 and IFIT3), which were clearly IRF9-dependent when we conduct the analysis as described using their RNA-seq data. Given these discrepancies, we avoided comparison. Instead, we discuss another published study³ using HeLa cells with wild-type, STAT1 KO, and

STAT2 KO lines treated with IFNB1 protein to identify STAT1- or STAT2-independent ISGs. We compared ISG induction in STAT2 KO cells with IRF9-independent ISGs. The comparative analysis is included in Supplementary Data 4 accessible via private Dryad link provided above. While IRF1 was returned as an IRF9-independent ISG from this analysis, most of the ISGs which passed the ≥ 2 -fold induction threshold by IFNB1 treatment and cancelation upon IFNAR2 KO were not induced in HeLa cells upon IFNB1 treatment. This reflects differences in cell lines, either because of epigenetic differences or off-target pathway activation following IFNAR signaling. This might indicate that after all, ISGs co-regulated across cell types might also be IRF9 dependent. In our study 78% of ISGs were IRF9-dependent, and only IRF1 and ICAM1 held up as IRF9-independent across cell types.

1. Ishikawa, H., and Barber, G.N. (2008). STING an Endoplasmic Reticulum Adaptor that Facilitates Innate Immune Signaling. *Nature* 455, 674–678. <https://doi.org/10.1038/nature07317>.

2. Zhang, L., Wei, X., Wang, Z., Liu, P., Hou, Y., Xu, Y., Su, H., Koci, M.D., Yin, H., and Zhang, C. (2023). NF- κ B activation enhances STING signaling by altering microtubule-mediated STING trafficking. *Cell Rep.* 42, 112185. <https://doi.org/10.1016/j.celrep.2023.112185>.

3. Urin, V., Shemesh, M., and Schreiber, G. (2019). CRISPR/Cas9-based Knockout Strategy Elucidates Components Essential for Type 1 Interferon Signaling in Human HeLa Cells. *J. Mol. Biol.* 431, 3324–3338. <https://doi.org/10.1016/j.jmb.2019.06.007>.

Point-by-point referee response

Reviewer #1 (Remarks to the Author):

This revised manuscript addresses the criticisms and suggestions from the prior review and presents a dataset and discussion that will be valuable for the scientific community. My only remaining suggestion relates to how the authors describe the relative importance of IRF3/7 direct gene induction versus IFN-stimulated (presumably STAT dependent) responses. The authors conclude "that an IFNB1 protein response is only leading into a "mild antiviral states" from a perspective of induction of direct acting antiviral genes, however, the concomitant upregulation of IRF7, IRF9 and other ISGs, including those involved in class I and class II antigen presentation, prepares cells for a possible infection and enables a stronger IRF7- and IRF9-assisted intracellular innate response and increased IFN gene expression." This statement is based on quantitative comparison of cGAMP v. IFN stimulated gene expression levels, but to conclude that physiologically relevant antiviral responses are primarily IRF3/7 driven after a largely priming role for IFN seems premature based on these experimental data. While there are well-documented feedforward pathways that regulate innate antiviral responses, to draw conclusions on the relative importance would require experiments in a more physiological setting, for instance, during a viral infection. Transfecting cells with cGAMP could bias the gene expression patterns that would be observed in a more biological context.

We have removed the corresponding statement in response to the reviewer's concern regarding potential overstatement of our conclusion based on comparative gene expression analysis of cGAMP- and IFNB1 protein-treated cell culture samples, particularly with respect to the relative contributions of IRF3- or IRF7-driven versus IFN-stimulated responses in establishing a physiologically relevant antiviral state. To address this point constructively and illustrate the relevance of our findings in a more physiologically meaningful context, we compiled and summarized the key gene groups identified in this study in Supplementary Data 5 (accessible via private Dryad link provided during previous revision) and demonstrated their applicability using Vesicular Stomatitis Virus (VSV) infection of THP1 cells as a representative biologically relevant example (Supplementary Figure 11). These revisions have been incorporated in the updated manuscript.

To improve clarity and facilitate easier interpretation for readers, we have also combined Figures 4 and 5 into a single Figure 4 and allocated a separate supplementary panel (Supplementary Figure 10B) for IRF9-independent ISGs in the revised version. The subsequent figures have been re-numbered accordingly.